# Anti-Fouling and Anti-Bacterial Modification of Poly(vinylidene fluoride) Membrane by Blending with the Capsaicin-Based Copolymer

**DOI:** 10.3390/polym11020323

**Published:** 2019-02-13

**Authors:** Xiang Shen, Peng Liu, Shubiao Xia, Jianjun Liu, Rui Wang, Hua Zhao, Qiuju Liu, Jiao Xu, Fan Wang

**Affiliations:** College of Chemistry and Environmental Science, Qujing Normal University, Qujing 655011, China; liupengxjlp@163.com (P.L.); xiashubiao401@163.com (S.X.); jianjun_liu@mail.qjnu.edu.cn (J.L.); 15969480853@163.com (R.W.); 13308749905@163.com (H.Z.); qj_liu@163.com (Q.L.); greengeny@163.com (J.X.); wfan321@126.com (F.W.)

**Keywords:** poly(vinylidene fluoride), biofouling, hydrophilicity, acrylomorpholine, capsaicin

## Abstract

Membrane fouling induced by the adsorption of organic matter, and adhesion and propagation of bacteria onto the surfaces, is the major obstacle for the wide application of membrane technology. In this work, the capsaicin-based copolymer (PMMA-PACMO-Capsaicin) was synthesized via radical copolymerization using methyl methacrylate (MMA), *N*-acrylomorpholine (ACMO) and 8-methyl-*N*-vanillyl-6-nonenamide (capsaicin) as monomers. Subsequently, the capsaicin-based copolymer was readily blended with PVDF to fabricate PVDF/PMMA-PACMO-Capsaicin flat sheet membrane via immersed phase inversion method. The effects of copolymer concentration on the structure and performance of resultant membranes were evaluated systematically. With increase of PMMA-PACMO-Capsaicin copolymer concentration in the casting solution, the sponge-like layer at the membrane cross-section transfers to macroviod, and the pore size and porosity of membranes increase remarkably. The adsorbed bovine serum albumin protein (BSA) amounts to PVDF/PMMA-PACMO-Capsaicin membranes decrease significantly because of the enhanced surface hydrophilicty. During the cycle filtration of pure water and BSA solution, the prepared PVDF/PMMA-PACMO-Capsaicin membranes have a higher flux recovery ratio (*FFR*) and lower irreversible membrane fouling ratio (*R_ir_*), as compared with pristine PVDF membrane. PVDF/PMMA-PACMO-Capsaicin membrane is found to suppress the growth and propagation of *Staphylococcus aureus* bacteria, achieving an anti-bacterial efficiency of 88.5%. These results confirm that the anti-fouling and anti-bacterial properties of PVDF membrane are enhanced obviously by blending with the PMMA-PACMO-Capsaicin copolymer.

## 1. Introduction

Poly(vinylidene fluoride) (PVDF) membrane is broadly utilized in municipal drinking water purification, wastewater treatment and food processing, owing to the excellent mechanical strength, chemical and thermal stability [1,2]. For example, the ultrafiltration PVDF membrane has been accepted as an advanced toolbox to effectively remove the organic matters (such as proteins, humic materials, polysaccharide and bacteria) in the wastewater steam [3,4,5]. However, because the PVDF membrane is inherently hydrophobic, the organic matters in the wastewater are easily adsorbed onto the membrane surface via chemical or physical interaction, resulting in the occurrence of membrane fouling. The membrane fouling has various disadvantages for the membrane performances, such as a shortness of service lifespan and the change of permeation and separation properties. Among the foulants, bacteria can exacerbate the membrane fouling. Once bacteria are permanently adhered to the surface, they can rapidly secrete organic compounds (proteins and polysaccharide, etc.), and then develop into a mature biofilm via the bacteria proliferation [6,7]. Consequently, a separation membrane with excellent fouling resistance is able not only to suppress the adsorption of organic compounds and bacteria, but also to prevent bacteria growth and propagation.

Hydrophilic modification of hydrophobic membrane is an effective strategy to alleviate the membrane fouling. Several hydrophilic polymeric materials have been introduced into the membrane to improve the surface hydrophilicity [8,9,10,11,12,13]. Among them, poly(ethylene glycol) (PEG) derivatives and zwitterionic polymers are the representative materials for the hydrophilic modification of membrane [10,11,12,13]. Carretier and coworkers [12] fabricated an anti-biofouling PVDF membrane by incorporating a tri-block copolymer (PEGMA_124_-*b*-PS_54_-*b*-PEGMA_124_) of poly(styrene) and poly(ethylene glycol) methacrylate. Li and coworkers [13] reported the surface grafting of amino-terminated hyperbranched polyglycerol (HPG-NH_2_) and zwtterionic poly(sulfobetaine methacrylate) (PSBMA) polymers onto PVDF membrane. All the prepared PEGylated and zwitterionic membranes were found to effective resist the protein adsorption and bacterial adhesion. Nevertheless, the solely hydrophilic modification of hydrophobic membrane may be not sufficient to inhibit the membrane biofouling, since the bacterial propagation and colonization are unavoidable once permanently adhesion of bacteria [14]. In the practical application, the synergistic effect of hydrophilic material and bactericidal agent is the ideal choice for the membrane modification.

Anti-bacterial agents can be classified into organic and inorganic materials. In the case of the membrane functionalized with inorganic materials, such as silver (Ag) [15,16,17], copper (Cu) [18,19] and zinc (Zn) [20,21], the release of these metal ions is inevitable and the long-term stability of anti-bacterial ability will be abated. The organic agent can be covalently immobilized onto the membranes to overcome the above-mentioned shortcomings. The organic anti-bacterial agents consist of polycation [22,23,24], halamine compounds [25,26], quaternary ammonium compound [27], and graphene oxide [28]. Among them, capsaicin-based materials are the derivatives and analogs of 8-methyl-*N*-vanillyl-6-nonenamide (Capsaicin) that is directly extracted from chili peppers. The derivatives and analogs of capsaicin have been reported to be efficient to inhibit the bacterial growth, and are thus used to construct the anti-fouling coatings [29]. Inspired by the excellent anti-bacterial ability, many capsaicin-based materials, including *N*-(5-methyl-3-isobutyl-2-hydroxy-benzyl)-acrylamide (MBHBA), (*N*-(4-hydroxy-3-methoxy-benzyl)-acrylamide (HMBA), *N*-(2-hydroxyl-3-methyl acrylamide-4,6-dimethyl benzyl) acrylamide (HMDA) and (5-methyl acrylamide-2,3,4 hydroxy benzyl) acrylamide (AMTHBA), are also introduced to improve the biofouling resistance of membranes [30,31,32,33,34,35]. For example, Gao and coworkers [35] reported the covalently grafting of MBHBA onto polysulfone (PSf) ultrafiltration membrane surface via UV-assisted graft polymerization. The modified PSf membrane exhibits anti-bacterial activity against *Escherichia coli* (*E. coli*). However, the surface hydrophilicity of the prepared membrane is not enough to meet the requirements in the practical application because of the hydrophobicity of capsaicin-based materials [36].

The present work aims to improve the anti-fouling and anti-bacterial properties of PVDF membrane by blending with a capsaicin-based copolymer. The copolymer PMMA-PACMO-Capsaicin was synthesized via radical copolymerization using methyl methacrylate (MMA), 8-methyl-*N*-vanillyl-6-nonenamide (capsaicin) and *N*-acryloylmorpholine (ACMO) as monomers. The synthesized copolymer exhibits the amphiphilic structure that consists of hydrophobic poly(methyl methacrylate) (PMMA) and hydrophilic poly (*N*-acryloylmorpholine) (PACMO) chains. After blending this copolymer into the membrane matrix, the PMMA chains are compatible with PVDF chains and thus ensure that the copolymer is not easily washed away in the filtration process. The introduction of amphiphilic copolymer was found to increase the membrane pore sizes during the membrane formation [34]. The membrane with a large pore size would suffer a significant flux reduction in a filtration process, and the mechanism was ascribed to the pore blockage followed by cake formation on the membrane surface [31]. Despite of the short-comings mentioned above, the hydrophilic PACMO chains within the synthesized PMMA-PACMO-Capsaicin copolymer can bind with the water molecules onto the membrane pore channel surface and membrane surface. The hydration layer resists the adsorption and adhesion of organic matters, thereby alleviating the formation of pore blockage and membrane fouling. In addition, the capsaicin moieties in the copolymer are expected to render the prepared membrane with durable anti-bacterial property. To the best of our knowledge, the anti-fouling and antibacterial modification of PVDF membrane using capsaicin-based copolymer as modifier are rarely investigated.

## 2. Materials and Methods

### 2.1. Materials

PVDF (Solef 1010, *M_w_* = 352,000, *M_w_*/*M_n_* = 2.3) was purchased from Solvey Co., Ltd., (Brussels, Belgium). Methyl methacrylate (MMA) and 8-methyl-*N*-vanillyl-6-nonenamide (Capsaicin) were supplied by Aladdin Co., Ltd., (Shanghai, China). Poly(ethylene glycol) (PEG, *M_w_* = 20,000), *N*-acryloylmorpholine (ACMO), 2,2′-azobisisobutyronitrile (AIBN), bovine serum albumin (BSA), *N*,*N*-dimethyl formamide (DMF) and *N*-methyl pyrrolidone (NMP) were bought from Xiyashiji Chemical Co., Ltd., (Shandong, China). All other chemicals, unless otherwise stated, were obtained from commercial sources and used as received.

### 2.2. Synthesis of Capsaicin-Based Copolymer

The capsaicin-based copolymer was synthesized via radical polymerization. In a three-necked flask, 16 g of MMA, 5.7 g of ACMO and 1 g of Capsaicin were dissolved into 50 mL of DMF. After bubbling with nitrogen gas for 15 min, 0.164 g of AIBN was added into the flask. The reaction solution was then bubbled with nitrogen gas for 10 min. Subsequently, the flask was sealed and the reaction was processed in an oil bath at 60 °C for 6 h. The polymer solution was precipitated into excess ethanol, and the product was isolated by filtration under vacuum. The product was then washed with pure water followed by filtration, and the process was repeated three times. Finally, the polymer sample was dried in an oven at 60 °C. In this work, the synthesized copolymer was denoted as PMMA-PACMO-Capsaicin.

The nuclear magnetic resonance proton spectrum (^1^H-NMR) of PMMA-PACMO-Capsaicin copolymer was investigated by an AV400 spectrometer (Bruker, Berlin, Germany) with deuterated dimethylsulfoxide (DMSO-*d*_6_) as the solvent and tetramethylsilane as the internal standard. Molecular weight and molecular weight distribution of PMMA-PACMO-Capsaicin copolymer were evaluated by a gel permeation chromatography (GPC) analysis system (Malvern Instruments Ltd., London, UK) tetrahydrofuran (THF) was used as the solvent and flow rate was set to 1 mL/min.

### 2.3. Membrane Preparation

PMMA-PACMO-Capsaicin was blended with PVDF to fabricate the PVDF/PMMA-PACMO-Capsaicin flat sheet membranes via the immersed phase inversion method. The compositions of casting solutions were shown in Table 1. To prepare casting solution, PVDF, PMMA-PACMO-Capsaicin and PEG were immersed into NMP solvent. After complete dissolution under stirring at 60 °C, the solution was degassed at 60 °C for 24 h. The casting solution was cooled to room temperature, and then spread onto a glass plate. The liquid membrane was cast using a casting knife with a thickness of 150 μm, and instantly immersed into a pure water bath (25 °C). After peeling off from the glass plate, the prepared PVDF/PMMA-PACMO-Capsaicin membranes were stored in pure water bath before use.

### 2.4. Membrane Characterization

The surface composition of membranes was determined by X-ray photoelectron spectroscopy (XPS, PHI5000C ESCA system, PHI Co., New York, NY, USA) using 300-W Al Kα radiation. The wide-scan XPS spectra were obtained at the pass energy of 160 eV, and the energy was set to 20 eV for high-resolution spectra. The binding energies were calibrated with respect to C–H component of C1s peak at 284.7 eV. The morphologies of membranes were observed on a scanning electron microscopy (SEM, Phenom ProX, Amsterdam, Netherlands) operating with the acceleration voltage of 5.0 kV. To obtain a clean break of membrane cross-sections, the membrane sample were fractured in liquid nitrogen. Subsequently, the samples were fixed to a standard stup using conductive tape. Atomic force microscopy (AFM, CSPM5500, Benyuannano, Beijing, China) was used to scan the membrane surface under tapping mode. The scanning area was set to 10 μm × 10 μm and the frequency was 1 Hz. The surface roughness parameters including the mean roughness (*R_a_*), root mean square (*R_q_*), and mean difference in the height between the highest peaks and the lowest valleys (*R_z_*) were obtained from AFM software. Mean pore size of membranes was measured by bubble-point method on a capillary flow porometry (3H-2000PB, BeiShiDe Instrument, Beijing, China).

The membrane porosity was investigated by the gravimetric method. The membrane sample was dried in a freezing dryer, and then immersed into isopropanol solution at room temperature for 24 h. The wet weight of membrane was measured after wiping away the excess isopropanol. The porosity of membrane was calculated from the relationship (1):
(1)$$\varepsilon =\frac{W_{1}-W_{2}}{A\times l\times \rho }\times 100\%$$
where *ε* is the porosity of membrane; *W*_1_ and *W*_2_ denote the wet and dry membrane weights, respectively. *A* is the membrane effective area (cm^2^), *ρ* is the isopropanol density (g/cm^3^), and *l* represents the membrane thickness (cm).

Contact angle of membrane surface was measured via an instrument (SCI4000D, Hengda Technology Co., Ltd., Beijing, China). Two microliters of pure water was dripped onto the membrane surface using a micro syringe. The contact angle values were determined by drop shape image analysis system.

BSA was chosen as a model protein to measure protein adsorption amounts of the membranes. The membrane samples were cut into small pieces (2 cm × 2 cm) and then transferred into BSA solution (0.5 g/L). After incubation at 25 °C for 24 h to reach adsorption equilibrium, the protein adsorption amounts were evaluated by measuring the concentration of BSA solution using a UV spectrophotometer (UV-1601, Shimadzu, Tokyo, Japan) at 280 nm. The reported value was the average of at least three measurements.

*Staphylococcus aureus* (*S. aureus*, ATCC6538) was used to determine the anti-bacterial activity of membranes. Aqueous suspensions of *S. aureus* were cultivated in 50 mL of yeast-dextrose broth (10 g of peptone, 8 g of beef extract, 5 g of NaCl, 5 g of glucose, and 3 g of yeast extract per liter of pure water) at 37 °C for 24 h. The S. aureus concentration was equivalent to ~10^9^ cells/mL on the basis of the estimation of optical density at 540 nm [37]. All the erlenmeyer flasks were sterilized and then 50 mL of yeast-dextrose broth was added into the flask. 0.1 mL of the bacteria suspension was pipetted out into the broth, and then about 100 mg of membrane sample was transferred to the bacteria suspension. After the membrane sample contacted with suspension at 37 °C for 4 h, 1 mL of the bacteria suspension in the flask was pipetted out, and then diluted with 9 mL of normal saline. The diluted suspension (0.1 mL) was spread onto a triplicate solid agar plate (with 9 cm of diameter). The plates were sealed and incubated at 37 °C for 24 h, and the numbers of the colonies on the plate were enumerated by the plate count method. A control experiment was also conducted without any membrane samples under the same conditions. The anti-bacterial activity (*Y*) was calculated as follows [34]:
(2)$$Y=\frac{N_{c}-N_{m}}{N_{c}}\times 100\%$$
where *N_m_* is the number of colony corresponding to the prepared membranes. *N_c_* denotes the number of colony in the control experiment. For each membrane sample, the reported value of anti-bacterial activity was the average of three repeated experiments.

### 2.5. Filtration Experiment

Filtration experiment was conducted using homemade cross-flow equipment in our laboratory. The pure water flux and BSA rejection were measured to characterize the permeation performance of the prepared membranes. The membrane sample was initially pressured with pure water at 0.2 MPa for at least 30 min. The pressure was decreased to 0.1 MPa, and then the steady pure water flux was calculated as follows:
(3)$$J=\frac{V}{A\times \Delta t}$$
where *V* is the volume of permeated pure water, (L); *A* represents the effective area of membrane, (m^2^), Δ*t* denotes the permeation time, (h). The BSA rejection (*R*) was calculated by the following equation:
(4)$$R=\left(1-\frac{C_{p}}{C_{f}}\right)\times 100\%$$
where *C_p_* and *C_f_* are the BSA concentration in the feed and permeated solution, respectively. The concentration was obtained from UV-spectrophotometer (UV-1601, Shimadzu, Tokyo, Japan) at a wavenumber of 280 nm.

To investigate the anti-fouling properties of the membranes in the permeation process, two-cycle filtration was performed with BSA as the foulant model. Each cycle filtration consisted of three steps, which were also conducted at 0.1 MPa. The first stage was the pure water filtration, and the flux was remarked as *J_w_*. The second step corresponded to the permeation process of BSA solution (1 g/L), and the flux of BSA solution was recorded as *J_B_*. At the third stage, the pure water filtration was performed again and the flux (*J_r_*) was also recorded, after the fouled membrane was washed with pure water for 30 min. During each cycle filtration, the irreversible membrane fouling ratio (*R_ir_*) caused by the firmly attachment of proteins was calculated by the following relationship:
(5)$$R_{ir}=\frac{J_{w}-J_{r}}{J_{w}}\times 100\%$$

The Flux recovery ratio (*FRR*) of the prepared membranes was defined as follows:
(6)$$FRR=\frac{J_{r}}{J_{w}}\times 100\%$$

## 3. Results

### 3.1. Characterization of PMMA-PACMO-Capsaicin Copolymer

The capsaicin-based copolymer (PMMA-PACMO-Capsaicin) was synthesized via radical copolymerization, and the chemical structure was investigated via 1H-NMR measurement. As shown in Figure 1, the chemical shift at a range of 0.73–0.93 ppm is attributed to –C–CH_2_– of MMA components in copolymer chains [38]. The presence of ACMO components can be ascertained by the peak at 2.69 ppm, which corresponds to –CH– in PACMO chains [39]. The peak at 8.84 ppm was ascribed to the –OH proton of phenol group in capsaicin components [34]. In addition, the chemical shift at 4.33 was assigned to the –CH_2_– proton that is adjacent to phenyl group of capsaicin [40]. The mole fraction of each component could be calculated using the integral areas of the characteristic peaks in the ^1^H-NMR spectrum. It is found that the mole fractions of MMA, ACMO and capsaicin components are 82.9%, 15.2% and 1.9%, respectively. The GPC analysis shows that the molecular weight (*M_w_*) is 52,160 g/mol, and the polydispersity index (PDI) is 3.65. These results indicate that the PMMA-PACMO-Capsaicin copolymer has been successfully synthesized.

### 3.2. Chemical Composition of Membrane Surfaces

X-ray photoelectron spectroscopy (XPS) analysis was carried out to evaluate the surface chemical composition of the prepared PVDF/PMMA-PACMO-Capsaicin membranes. Figure 2 shows the wide-scan XPS spectra of membrane surfaces. In the case of pristine PVDF membrane (M0), the surface exhibits major emission peaks at 688.0 eV for F1s and 291.0 eV for C1s. A small emission at 532.9 eV is also observed on PVDF membrane surface, which may be ascribed to the adsorption of oxygen molecules [41]. Compared with the pristine PVDF membrane, a new peak at 400.2 eV is attributed to the N1s signal in the XPS spectra of PVDF/PMMA-PACMO-Capsaicin membranes (M1, M2 and M3). The mole percentages of surface elements were listed in Table 2. With the increase of PMMA-PACMO-Capsaicin copolymer concentration, the N1s and O1s percentages determined via XPS increase from 0 to 3.32% and from 2.82% to 13.46%, respectively. The peak compositions of C1s peak in the XPS spectrum were further investigated, and the results were also shown in Figure 2. The components at 290.6 eV for CF_2_ and 286.3 eV for CH_2_ are observed in the C1s core-level spectrum of pristine PVDF membrane (M0). In addition, the small peak at 284.7 eV is ascribed to the neutral C-H groups, which are resulted from the branching sites and end groups within PVDF chains. In the case of PVDF/PMMA-PACMO-Capsaicin membranes, the C1s spectra can be divided into five components. Except for the peaks of the PVDF chains, the components with the binding energies of 289.2 eV and 288.5 eV are ascribed to O–C=O and C=O groups [23,42], which is originated from PMMA and PACMO chains of copolymer, respectively. Due to the peaks of C-N, C–OH, C–O–C and CH_2_ have the same binding energies, they are integrated as a sole component at 286.3 eV. As shown in Table 2, the percentages of O–C=O are 4.26%, 7.15% and 8.09% for M1, M2 and M3, and the fractions of C=O peaks are 1.44%, 2.67% and 3.19% for M1, M2 and M3, respectively. These results show that the capsaicin-based copolymer has been blended into the membranes. On the basis of the elemental percentage on the membrane surface, the N/F fractions were readily calculated. It can be seen from Table 2 that the experimental value of N/F is much higher than theoretical value, which confirms that the PACMO chains and capsaicin moieties are easily segregated onto the membrane surface or pore channel surface during the membrane formation process [39].

### 3.3. Membrane Morphologies

The surface and cross-sectional structures of membranes were observed via SEM. As viewed in Figure 3, the prepared membranes exhibit an asymmetric cross-sections consisting of a skin layer and a porous sub-layer. In the case of pristine PVDF membrane (M0), the cross-section is mainly occupied by the macroviod structure and the sponge-like layer is observed near the bottom surface of membrane. After the introduction of PMMA-PACMO-Capsaicin copolymer, the sponge-like structure disappears and then completely changes into macroviods, which is revealed in cross-sections of PVDF/PMMA-PACMO-Capsaicin membranes (M1, M2 and M3). Figure 3 also shows that the pristine PVDF membrane has a dense surface. With the addition of PMMA-PACMO-Capsaicin copolymer, the pore structure is gradually formed on the surface of PVDF/PMMA-PACMO-Capsaicin membranes (M1, M2 and M3). With the increase of PMMA-PACMO-Capsaicin concentration in the casting solution, the surface porous structures become pronounced. The pore size and bulk porosity of membranes were measured and the results were summarized in Table 3. It can be seen that the porosity *(ε*) and mean pore size (*r_m_*) increase from 68.6% to 75.3% and from 100.5 nm to 179.2 nm, respectively. This result is also consistent with the surface and cross-sectional morphologies of membranes. The typical morphology of membranes is ascribed to the occurrence of instantaneous liquid–liquid de-mixing process when the nascent membrane is immersed into coagulant (water) [43]. Since the amphiphilic PMMA-PACMO-Capsaicin copolymer is introduced into the casting solution, the hydrophilic PACMO chains from the copolymer move towards the interface between water and casting solution. The presence of hydrophilic PACMO chains will increase the affinity of the coagulant (water) and the casting solution. The exchange rate between solvent and non-solvent is significantly accelerated, which is expected to facilitate the liquid–liquid de-mixing process.

The surface microstructure of membranes was further scanned via AFM, and the AFM images with the area of 10 μm × 10 μm were illustrated in Figure 4. AFM images suggest that the prepared membrane surfaces are not smooth, where the surface contains several kinds of valleys and peaks. The surface roughness parameters of membranes, in terms of mean roughness (*R_a_*), root mean square roughness (*R_q_*) and mean height difference (*R_z_*) between the highest peaks and the lowest depressions, were also listed in Table 3. The surface roughness parameters of the membranes become greater with higher feed concentration of PMMA-PACMO-Capsaicin copolymer in the casting solution. This result may be explained that more PACMO chains are segregated at the membranes surface with the increment of copolymer concentration [43]. In addition, the surface roughness strongly depends on the valleys characterizing surface pores and peaks characterizing the nodules. When the deep valleys and high peaks are formed on the membrane surface, the high values of roughness parameters are obtained [44]. In this work, the high surface pore size is generated from the fast liquid–liquid de-mixing process of casting solution during the membrane formation, which also renders the membrane with rough surface structure.

### 3.4. Hydrophilicity of Membranes

The surface hydrophilicity of membranes was evaluated via water contact angle measurements. Figure 5 shows the water contact angles of the prepared membranes. The water contact angle of pristine PVDF membrane is as high as 88.6° because of the low hydrophilicity. With the increasing concentration of PMMA-PACMO-Capsaicin copolymer in the casing solution, the water contact angle of the prepared membranes is significantly reduced. Especially for PVDF/PMMA-PACMO-Capsaicin membrane M3, the contact angle is 57.8°, about 31° decrease comparing with the pristine PVDF membrane. This difference is ascribed to the fact that the introduction of copolymer results in the large pore size of membrane surface. When the water droplet contacts with the membrane surface, the water molecules are easily diffused into the membrane pores. Besides, the hydrophilic PACMO chains tend to segregate to the membrane pore surface during membrane formation process. The strong interaction forces induced by hydrogen bonds facilitate the quick spreading of water molecules [39]. This result indicates that the incorporation of PMMA-PACMO-Capsaicin copolymer enhances the surface hydrophilicity of membranes obviously.

### 3.5. Permeation and Separation Properties of Membranes

The permeation and separation properties of the prepared membranes were investigated via pure water and BSA solution (1 g/L) filtration, respectively. The pure water fluxes and BSA rejections of membranes were measured and the average value of at least six measurements was reported in Table 4. It can be seen that the pristine PVDF membrane (M0) has the pure water flux of 149.2 L/m^2^h under the operational pressure of 0.1 MPa. The pure water flux of membranes increases obviously with the addition of PMMA-PACMO-Capsaicin copolymer. In particular, the water flux of PVDF/PMMA-PACMO-Capsaicin membrane M3 is much higher than that of pristine PVDF membrane (M0). However, BSA rejections of the prepared membranes decrease with the increase of PMMA-PACMO-Capsaicin copolymer. The BSA rejection of PVDF/PMMA-PACMO-Capsaicin membrane M3 is only 67.6% of pristine PVDF membrane, achieving a value of 64.6%. It is obvious that there is a trade-off between the permeation and separation properties for the prepared membranes. Similar experimental results were also found in the previous work [34]. This typical behavior is attributed to the surface hydrophilicity, membrane porosity and pore size. The enhanced surface hydrophilicity and formation of the membrane structure with a large pore size and porosity are the synergistic factors for increasing the water fluxes of membranes. On the other hand, the porous membrane structures also allow the BSA molecules to permeate through the membrane pores, and the BSA solutes are not effectively retained. These results are also consistent with the SEM images and water contact angle measurement of membranes.

### 3.6. Anti-Fouling Properties of Membranes

The anti-fouling property of membranes was evaluated via a static protein adsorption measurement using BSA as model protein. Figure 6 shows the adsorbed BSA amounts of the prepared membranes. As the pristine PVDF membrane (M0) is relatively hydrophobic, the BSA molecules are easily adhered and the amount reaches to 77.6 μg/cm^2^. The amounts adhered to PVDF/PMMA-PACMO-Capsaicin membranes (M1, M2 and M3) are lower than that to pristine PVDF membrane, and BSA adsorption decreases with the increasing concentration of PMMA-PACMO-Capsaicin copolymer. Especially for PVDF/PMMA-PACMO-Capsaicin membrane M3, the BSA protein adsorption reduces to 27.6 μg/cm^2^, about 35.6% of BSA adsorption to the PVDF membrane. The improvement of protein fouling resistance may be attributed to the hydration interaction between water molecules and hydrophilic PACMO chains, resulting in a repulsive force of membrane surface to protein molecules [39,42].

To investigate the anti-fouling properties of membranes in the separation process, two-cycle filtration was performed at 0.1 MPa. Each cycle includes three phases: (1) the filtration process of pure water, (2) the permeation of BSA solution (1 g/L) and (3) the repeat water filtration after the fouled membrane is flushed with pure water for 30 min. Figure 7 shows the time-dependent flux curves of the prepared membranes. It is found that the flux values of membranes increase with the increasing content of PMMA-PACMO-Capsaicin copolymer. Compared with filtration phase of pure water, due to the formation of cake layer onto the membranes, the flux values of BSA solution reduce remarkably. The membrane fouling during the cycle filtration was quantified using the irreversible membrane fouling ratio (*R_ir_*). From Figure 8, the *R_ir_* decreases with increasing PMMA-PACMO-Capsaicin concentration. Actually, the relative reduction of pure water flux is attributed to the combined effects of adsorption and deposition of protein molecules during permeation process [45]. The decrease of *R_ir_* suggests that incorporation of PMMA-PACMO-Capsaicin has significantly suppressed the irreversible adsorption and deposition effects of BSA proteins in the filtration process. Figure 8 also shows pure flux recovery ratios (*FRR*) of the prepared membranes. The *FRR* values of PVDF/PMMA-PACMO-Capsaicin membranes (M1, M2 and M3) are much higher than that of pristine PVDF membrane (M0). As for PVDF/PMMA-PACMO-Capsaicin membranes, the improvement in surface hydrophilicity effectively decreases the interaction force between membrane surface and BSA solute. The BSA molecules are not firmly adhered to the PVDF/PMMA-PACMO-Capsaicin membranes, as revealed in protein adsorption measurement. After the fouled membrane is washed with pure water, the pure water flux can be recovered to a high level. These results confirm that the prepared PVDF/PMMA-PACMO-Capsaicin membranes have the excellent anti-fouling properties in the separation process.

### 3.7. Anti-Bacterial Properties of Membranes

To investigate the anti-bacterial activity of the prepared PVDF/PMMA-PACMO-Capsaicin membranes, the anti-bacterial tests against *S. aureus* were carried out. The membrane sample was contacted with *S. aureus* suspension at 37 °C for 4 h, and then was taken out from the suspension. The rest suspension was diluted, and then spread onto the solid culture medium to incubate at 37 °C for another 24 h. Figure 9 shows the images of petri dishes after *S. aureus* growth. Compared with the blank sample, many bacteria colonies are observed after the pristine PVDF membrane (M0) contacts with *S. aureus*, indicating the inactivity of PVDF membrane against bacteria. However, after the addition of PMMA-PACMO-Capsaicin copolymer, number of bacteria colonies obviously decreases. Based on the number of bacteria colonies on petri dishes, anti-bacterial efficiency of the prepared PVDF/PMMA-PACMO-Capsaicin membranes was calculated using the procedures described in the previous work [34], and the result was shown in Table 5. It can be seen that the pristine PVDF membrane (M0) has the minimal anti-bacterial activity. With the increasing concentration of PMMA-PACMO-Capsaicin copolymer, the bactericidal efficiency of the prepared membranes increases dramatically from 1.5% to 88.5%. This result confirms that the prepared PVDF/PMMA-PACMO-Capsaicin membranes have the excellent capability to inhibit the bacterial growth.

## 4. Conclusions

The capsaicin-based copolymer of PMMA-PACMO-Capsaicin was synthesized via radical copolymerization, and then was directly blended to fabricate PVDF/PMMA-PACMO-Capsaicin flat sheet membrane via immersed phase inversion method. During formation process, PACMO chains and capsaicin moieties within copolymer structure tended to segregate onto the membrane surface. With the increase of PMMA-PACMO-Capsaicin concentration in the casting solution, the sponge-like structure at the cross-sections gradually transferred into macroviod. The incorporation of copolymer resulted in the remarkable increment of pore size and porosity, and the formation of rough membrane surface. Due to the improvement in surface hydrophilicty, the adsorbed BSA amounts to PVDF/PMMA-PACMO-Capsaicin membranes were smaller than that of the pristine PVDF membrane. The pure water fluxes of the prepared PVDF/PMMA-PACMO-Capsaicin membranes increased, whereas the BSA rejection values decreased with the addition of PMMA-PACMO-Capsaicin copolymer. In the cycle filtration process, the prepared PVDF/PMMA-PACMO-Capsaicin membranes had a higher flux recovery ratio (*FFR*) and a smaller relative flux reduction (*RFR*), as compared with pristine PVDF membrane. The bactericidal efficiency of the prepared membrane was as high as 88.5%, indicating an excellent bio-fouling resistance. Consequently, outcomes of the present work show that the synthesized capsaicin-based copolymer can be candidate for the preparation and design of the anti-fouling and anti-bacterial PVDF membranes.

## Figures and Tables

**Figure 1 polymers-11-00323-f001:**
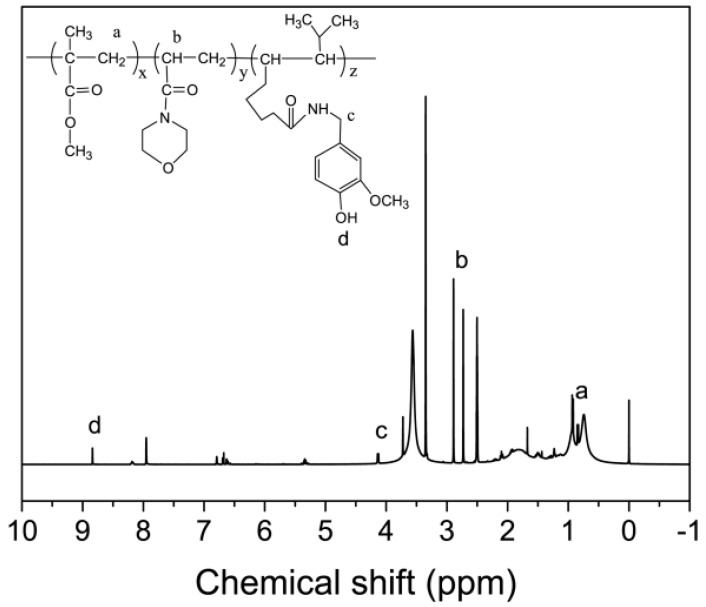
^1^H-NMR spectrum of PMMA-PACMO-Capsaicin copolymer.

**Figure 2 polymers-11-00323-f002:**
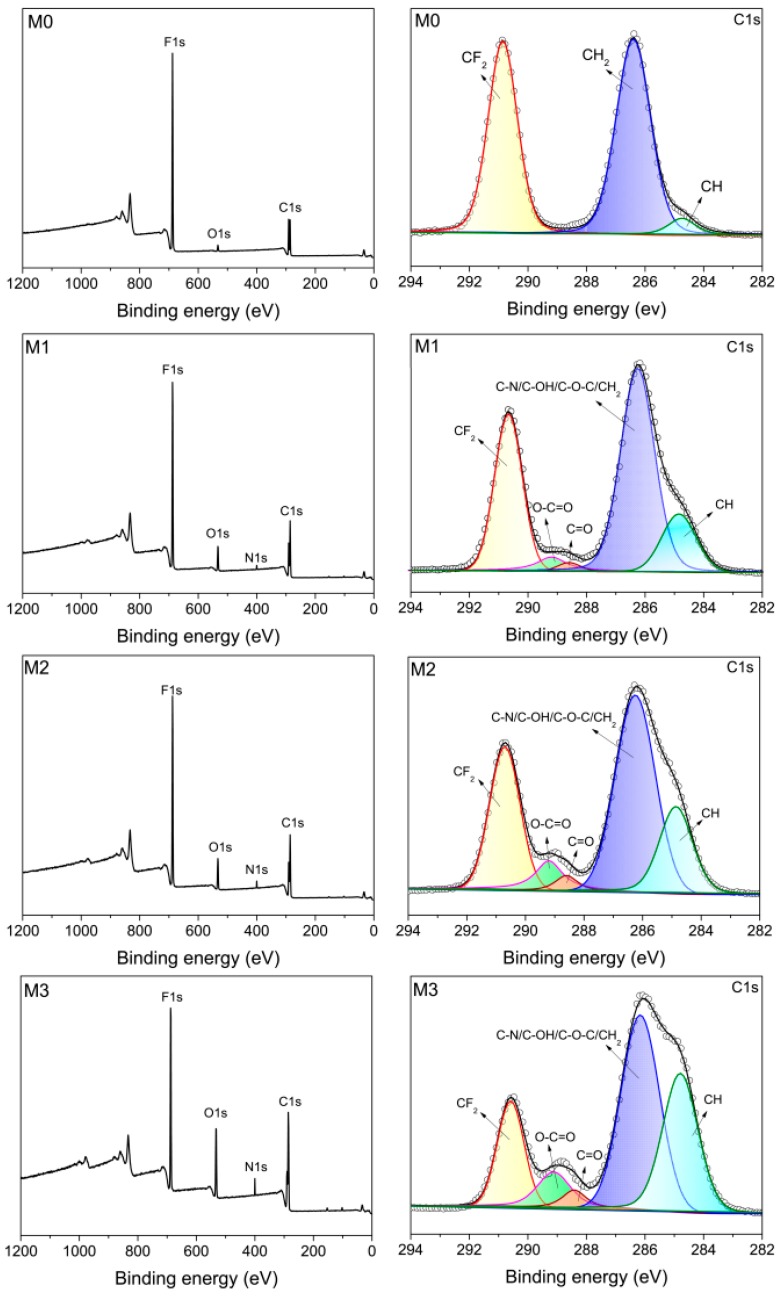
Wide-scan and C1s XPS spectra of membrane surfaces.

**Figure 3 polymers-11-00323-f003:**
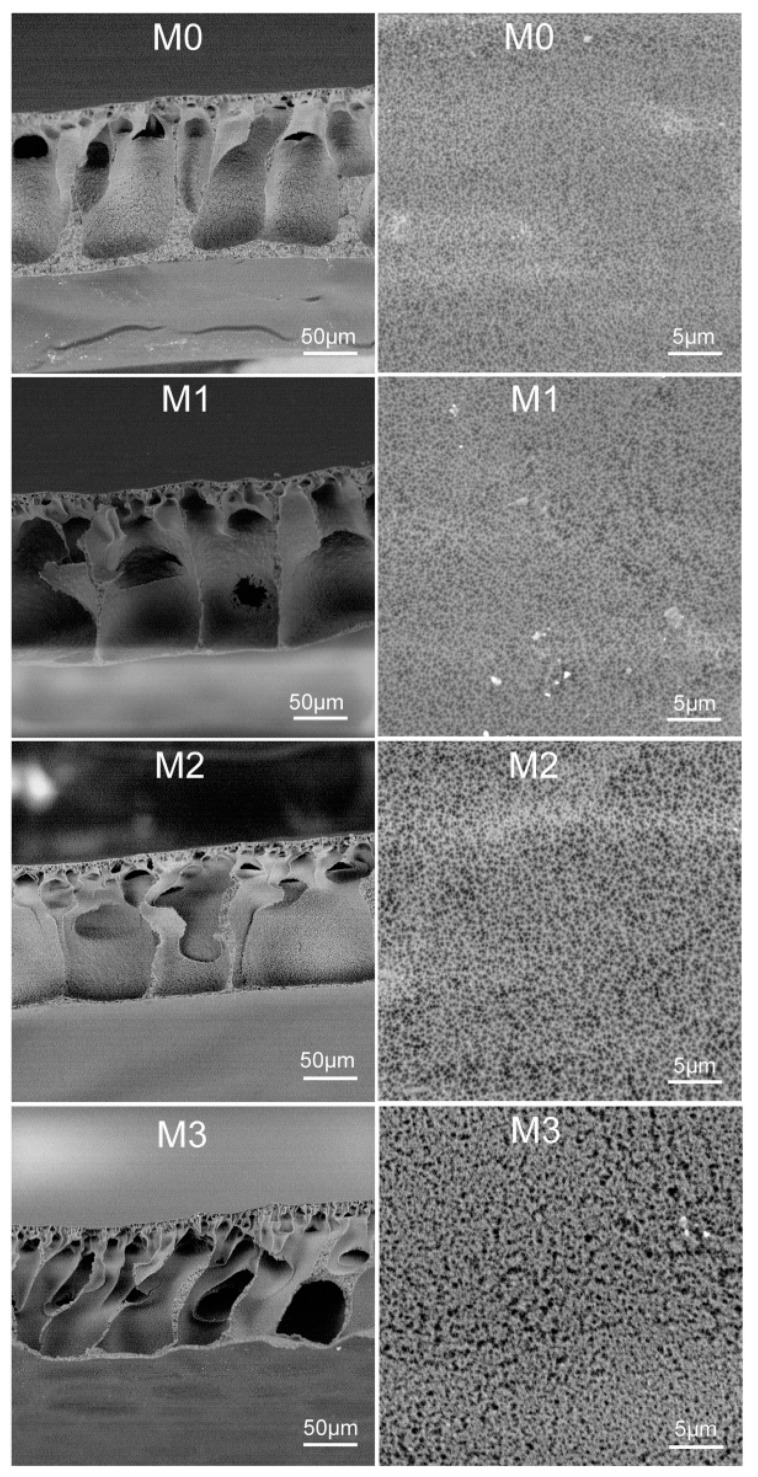
Cross-sectional and surface structures of membranes.

**Figure 4 polymers-11-00323-f004:**
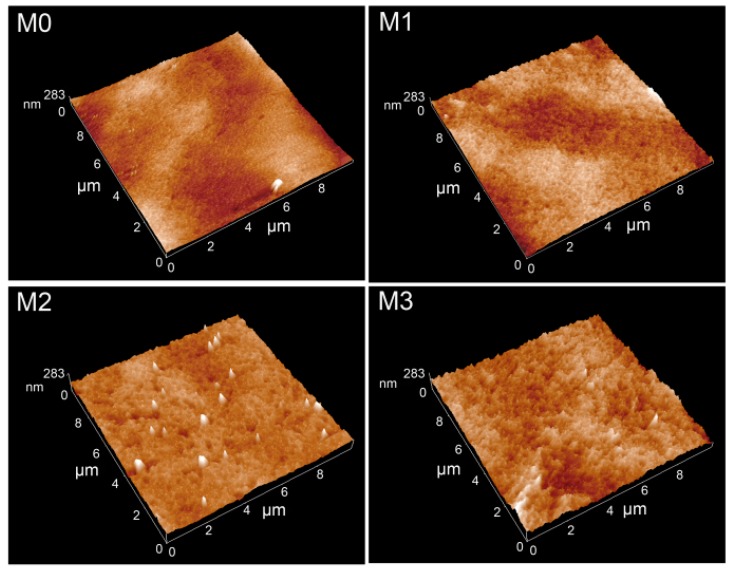
AFM images of membrane surfaces.

**Figure 5 polymers-11-00323-f005:**
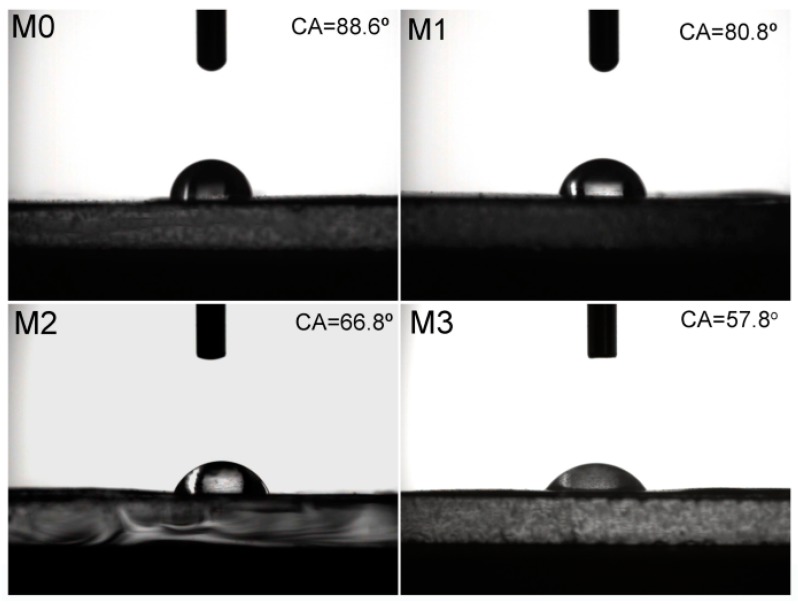
Water contact angles of membranes.

**Figure 6 polymers-11-00323-f006:**
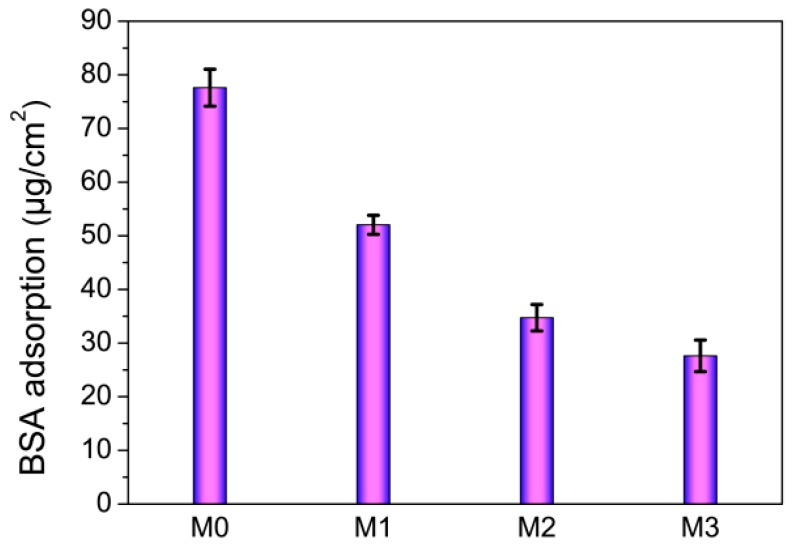
Amounts of protein adsorbed onto the prepared membranes.

**Figure 7 polymers-11-00323-f007:**
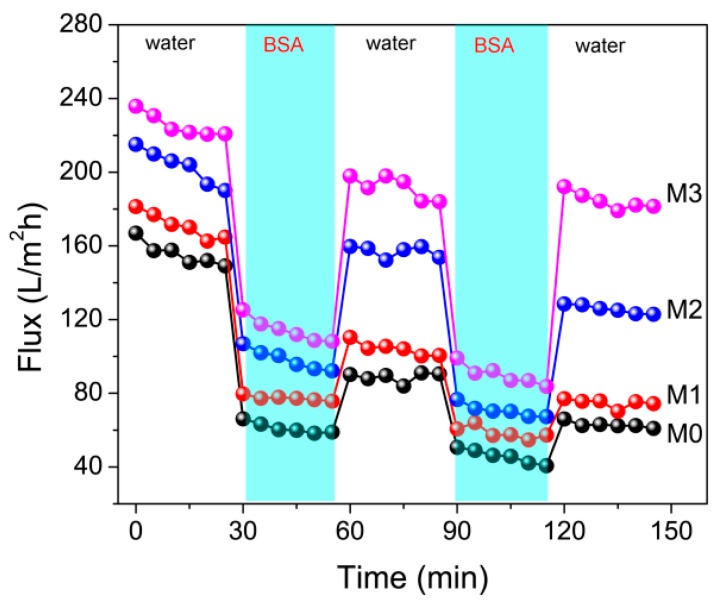
Cycle filtrations of membranes. The each cycle includes three steps: Pure water filtration, BSA solution (1 g/L) filtration and pure water filtration process after water flushing.

**Figure 8 polymers-11-00323-f008:**
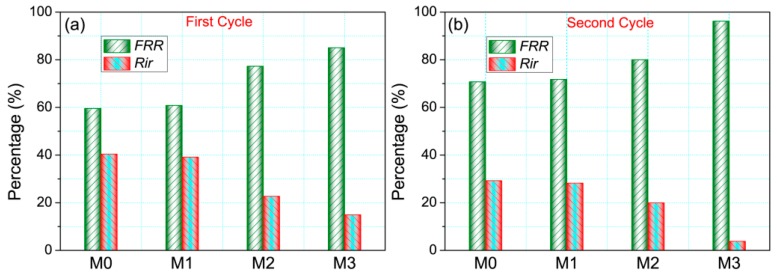
Irreversible membrane fouling ratio (*R_ir_*) and flux recovery ratio (*FRR*) of membranes in the cycle filtration. (**a**,**b**) represent the first and second cycle filtration, respectively.

**Figure 9 polymers-11-00323-f009:**
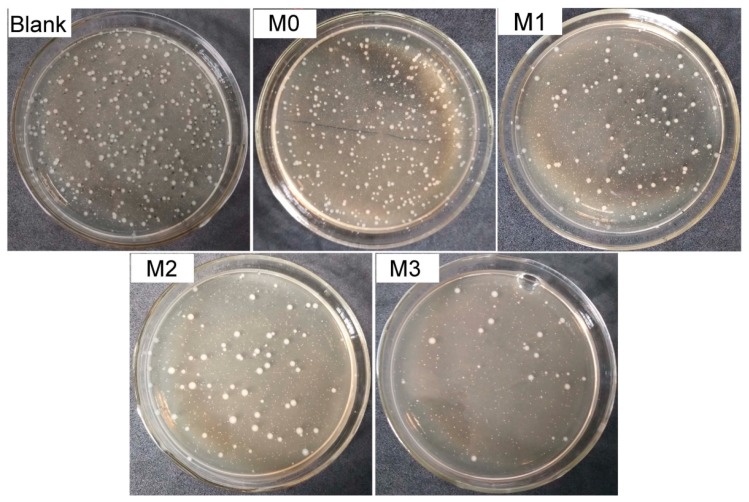
Photographs of the prepared membranes against *S. aureus* growth.

**Table 1 polymers-11-00323-t001:** The detailed compositions of casting solutions.

Membrane Sample	PVDF (wt%)	PMMA-PACMO-Capsaicin (wt%)	PEG (wt%)	NMP (wt%)
M0	15.0	0.0	2.5	82.5
M1	12.5	2.5	2.5	82.5
M2	10.0	5.0	2.5	82.5
M3	7.5	7.5	2.5	82.5

**Table 2 polymers-11-00323-t002:** Chemical compositions of membrane surfaces.

Sample	Chemical Elements (%)				(N/F)*_e_*(%)	(N/F)*_t_*(%)	C 1s (%)				
	C1s	F1s	O1s	N1s			CF_2_	O–C=O	C=O	CH	C–N/C–OH/C–O–C/CH_2_
M0	52.66	44.52	2.82	—	—	—	44.01	—	—	4.27	51.72
M1	55.53	35.26	7.50	1.71	4.85	0.98	29.68	4.26	1.44	13.75	50.87
M2	57.77	30.08	9.57	2.58	8.58	2.45	25.45	7.15	2.67	18.66	46.07
M3	60.95	22.27	13.46	3.32	14.91	4.97	16.07	8.09	3.19	29.87	42.78

*e* experimental value, *t* theoretical value.

**Table 3 polymers-11-00323-t003:** Roughness parameters, porosity and pore sizes of hybrid membranes.

Sample	*R_a_* (nm)	*R_q_* (nm)	*R**_z_*(nm)	*ε* (%)	*r_m_* (nm)
M0	14.3 ± 2.1	17.3 ± 3.2	160.5 ± 6.4	68.6	100.5
M1	16.4 ± 3.6	20.1 ± 3.4	170.5 ± 14.4	69.2	146.1
M2	15.8 ± 2.3	19.1 ± 1.6	204.0 ± 11.9	72.5	164.6
M3	19.4 ± 1.7	25.4 ± 2.0	231.0 ± 18.1	75.3	179.2

**Table 4 polymers-11-00323-t004:** Pure water fluxes and BSA rejections of the prepared membranes.

Sample	Pure Water Flux (L/m^2^h)	BSA Rejection (%)
M0	149.2	95.6
M1	171.3	83.4
M2	203.1	71.3
M3	225.5	64.6

**Table 5 polymers-11-00323-t005:** Anti-bacterial efficiency of the prepared membranes.

Sample	Blank	M0	M1	M2	M3
Number of bacteria colonies	260	256	89	74	30
Anti-bacterial efficiency (%)	0	1.5	65.8	71.5	88.5

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
