# Peer review of "Anti-Fouling and Anti-Bacterial Modification of Poly(vinylidene fluoride) Membrane by Blending with the Capsaicin-Based Copolymer"

_polymers, 2019, doi:10.3390/polym11020323_

Round 1
Reviewer 1 Report
SUMMARY
The authors have explored methods to form a co-polymer system including capsaicin and PVDF in order to enhance hydrophilicity and improve bactericidal properties with the eventual aim of use as antifouling membrane components. The paper may be of interest to those in the functional polymers and filter membranes fields, amongst others. In general, the paper reads well but there are some minor spelling and grammatical mistakes that need to be corrected. Also, in certain cases, the writing is a little unclear. With the inclusion of the recommended changes below, the paper will be suitable for publication.
ABSTRACT:
- Some parts of the abstract are clumsily/unclearly phrased and the flow is hard to follow
- Perhaps be specific about bacteria, test methods etc.
INTRODUCTION:
- Perhaps also make reference to the importance of pore sizes on fouling etc., for membranes
METHODOLOGY:
- NMR acquisition details? Pulse times? Number of data points?
- XPS acquisition parameters? Pass energies for acquisition? Step sizes? Calibration standard used?
- SEM acquisition parameters? Accelerating voltage? Sample preparation/coating?
- AFM acquisition parameters? What mode?
RESULTS & DISCUSSION:
- The term ‘about’ is a little unscientific and imprecise, yet it is used frequently in the manuscript. Generally in the case of experimental lab work and specifically for microbiology, it is a little too conversational
- For contact angle data, values less than 90° are considered hydrophilic by most common definitions. As such, it seems incorrect to call the samples hydrophobic. I would strongly recommend looking at this again.
- The antimicrobial data analysis seems only partly completed/analysed. The standard processes is to convert the data to the relevant log value. Furthermore, it is usually convention to apply a relevant statistical test in order to ascertain the actual antimicrobial efficacy of the synthesised samples. As such, whilst the data looks impressive at first glance, it is difficult to truly appraise the system.
CONCLUSION:
- The claims made and findings reported seem fine.
Author Response
Point 1: Comments and Suggestions for Authors, The authors have explored methods to form a co-polymer system including capsaicin and PVDF in order to enhance hydrophilicity and improve bactericidal properties with the eventual aim of use as antifouling membrane components. The paper may be of interest to those in the functional polymers and filter membranes fields, amongst others. In general, the paper reads well but there are some minor spelling and grammatical mistakes that need to be corrected. Also, in certain cases, the writing is a little unclear. With the inclusion of the recommended changes below, the paper will be suitable for publication.
Response 1: Thanks for your comments on our manuscript. This work is aimed to increase the surface hydrophilicity and anti-bacterial ability of PVDF membrane by incorporation the synthesized capsaicin-based copolymer. In the revised paper, we have taken the great efforts to correct the spelling and grammatical mistakes. In addition, the sentences are logically modified to reach the meaning of expression. All the changes are illustrated in the revised paper using “Track Changes”. Please check it!
Point 2: ABSTRACT: Some parts of the abstract are clumsily/unclearly phrased and the flow is hard to follow. Perhaps be specific about bacteria, test methods etc.
Response 2: the Abstract section has been modified in the revised paper. The sentences ‘The incorporation of PMMA-PACMO-Capsaicin leads to the change of membrane structure, in which the sponge-like layer at the membrane cross-section transfers to macroviod, and the pore size and porosity of membranes increase remarkably.’ has been changed to ‘With increase of PMMA-PACMO-Capsaicin copolymer concentration in the casting solution, the sponge-like layer at the membrane cross-section transfers to macroviod, and the pore size and porosity of membranes increase remarkably.’. In addition, the specifications on the bacterial test and cycle filtration measurements were provided. ‘During the cycle filtration…..’ was changed into ‘of pure water and BSA solution….’. ‘……is found to suppress the bacterial growth……’ was changed into ‘is found to suppress the growth and propagation of Staphylococcus aureus bacteria……’. The sentence “These results confirm that the anti-fouling and anti-bacterial properties are enhanced obviously by blending capsaicin-based copolymer into the PVDF membrane.” was rewritten into “These results confirm that the anti-fouling and anti-bacterial properties of PVDF membrane are enhanced obviously by blending with the PMMA-PACMO-Capsaicin copolymer.” in lines 30 and 31.
Point 3: INTRODUCTION: Perhaps also make reference to the importance of pore sizes on fouling etc., for membranes
Response 3: The membrane pore size is one of the key factors affecting membrane fouling. The larger membrane pore size always results in a significant membrane fouling due to the pore blockage followed by cake formation. In the revised paper, the sentences “The introduction of amphiphilic copolymer was found to increase the membrane pore sizes during the membrane formation[35]. The membrane with a large pore size would suffer a significant flux reduction in a filtration process, and the mechanism was ascribed to the pore blockage followed by cake formation on the membrane surface[32]. Despite of the short-comings mentioned above, the hydrophilic PACMO chains within the synthesized PMMA-PACMO-Capsaicin copolymer can bind with the water molecules onto the membrane pore channel surface and membrane surface. The hydration layer resists the adsorption and adhesion of organic matters, thereby alleviating the formation of pore blockage and membrane fouling. In addition, the capsaicin moieties in the copolymer are expected to render the prepared membrane with durable anti-bacterial property.”have been inserted on page 3.
Point 4: METHODOLOGY: NMR acquisition details? Pulse times? Number of data points? XPS acquisition parameters? Pass energies for acquisition? Step sizes? Calibration standard used? SEM acquisition parameters? Accelerating voltage? Sample preparation/coating? AFM acquisition parameters? What mode?
Response 4: In this work, NMR was used to investigate the chemical structure and compositions of copolymer. XPS was utilized to characterize the surface compositions of membranes. The SEM and AFM were used to observe the membrane structures. In the revised paper, the details on these measurements were provided. “The wide-scan XPS spectra were obtained at the pass energy of 160 eV, and the energy was set to 20 eV for high-resolution spectra. The binding energies were calibrated with respect to C-H component of C1s peak at 284.7 eV.” were inserted at lines 165-167. “….operating with the acceleration voltage of 5.0 kV. To obtain a clean break of membrane cross-sections, the membrane sample were fractured in liquid nitrogen. Subsequently, the samples were fixed to a standard stup using conductive tape.” were inserted at lines 168-170. “Atomic force microscopy (AFM, CSPM5500, Benyuannano, China) was used to scan the membrane surface under tapping mode. The scanning area was set to 10 μm × 10 μm and the frequency was 1 Hz.” was added in lines 171-172.
Point 5: The term ‘about’ is a little unscientific and imprecise, yet it is used frequently in the manuscript. Generally in the case of experimental lab work and specifically for microbiology, it is a little too conversational
Response 5: Thanks for your advices. In the revised paper, the terms on ‘about’ have been modified to meet the scientific and precise requirements, which are shown as ‘Track Changes’ mode.
Point 6: For contact angle data, values less than 90° are considered hydrophilic by most common definitions. As such, it seems incorrect to call the samples hydrophobic. I would strongly recommend looking at this again.
Response 6: Thanks for your comments. In this work, the water contact angle of pristine PVDF membrane was 88.6° that was smaller than 90°, indicating that the surface of PVDF membrane is relatively hydrophilic. In the revision, the term on ‘hydrophobic’ has been corrected. ‘…because of its hydrophobic nature’ was changed into ‘…because of the low hydrophilicity’.
Point 7: The antimicrobial data analysis seems only partly completed/analysed. The standard processes is to convert the data to the relevant log value. Furthermore, it is usually convention to apply a relevant statistical test in order to ascertain the actual antimicrobial efficacy of the synthesised samples. As such, whilst the data looks impressive at first glance, it is difficult to truly appraise the system.
Response 7: In this work, the antibacterial test was conducted to measure the anti-bacterial efficiency of prepared membranes. The procedures for this measurement were similar with the previous works [Journal of Membrane Science 445 (2013) 146–155; Ind. Eng. Chem. Res. 2015, 54, 11312−11318]. The anti-bacterial efficiency of membranes were calculated based on the number of colonies using the equation: Y=[(Nc-Nm)/Nc]×100%, where Nm is the number of colony corresponding to the prepared membranes. Nc denotes the number of colony in the control experiment. For each membrane sample, the reported value of anti-bacterial activity was the average of three repeated experiments. To express the accuracy of anti-bacterial efficiency, the relevant references were cited in the revise paper in lines 212 and 433.

Reviewer 2 Report
This study investigates a novel approach to improve the antifouling and antibacterial ability of Poly(vinylidene fluoride) membranes. Their approach involves blending Poly(vinylidene fluoride) with a copolymer of methyl methacrylate, acrylomorpholine and 8-methyl-N-vanillyl-6-nonenamide (capsaicin) . The effect of different concentrations of the copolymers on the composition, structure such as pore size, porosity, hydrophobicity was evaluated to form an understanding of the material. The authors then use that as a basis and also investigate the functional behaviors of the different membranes in terms of flux, protein adsorption, membrane fouling and antibacterial effects.
This is a well thought out study with a clear hypothesis and a systematic investigation of the chemical, physical and functional aspects of the capsaicin-based copolymer membranes. The manuscript is well written and the tables and figures are consistent and clear.
The authors may want to verify if they meant ‘advantages’ or ‘disadvantages’ in line 41 of their introduction.
I do not have any further broad comments to the authors since the authors have presented a detailed study where their data supports the hypothesis and the paper is recommended for publication in the current form.
Author Response
Point 1: This study investigates a novel approach to improve the antifouling and antibacterial ability of Poly(vinylidene fluoride) membranes. Their approach involves blending Poly(vinylidene fluoride) with a copolymer of methyl methacrylate, acrylomorpholine and 8-methyl-N-vanillyl-6-nonenamide (capsaicin) . The effect of different concentrations of the copolymers on the composition, structure such as pore size, porosity, hydrophobicity was evaluated to form an understanding of the material. The authors then use that as a basis and also investigate the functional behaviors of the different membranes in terms of flux, protein adsorption, membrane fouling and antibacterial effects.
Response 1: Thanks for your positive comments on our manuscript. In this work, the capsaicin-based copolymer (PMMA-PACMO-Capsaicin) was synthesized and blended with PVDF to improve the anti-fouling and anti-bacterial abilities. During membrane formation, the hydrophobic PMMA segments are compatible with PVDF and ensure that the amphiphilic copolymer is not easily washed away from the membrane, while the hydrophilic PACMO chains will endow the membrane with the desirable hydrophilicity and anti-fouling property through the surface segregation during the phase inversion process. In addition, the presence of Capsaicin is expected to improve the anti-bacterial efficiency of membranes. The effects of copolymer concentration on the structure and performance were investigated by flux, protein adsorption, cycle filtration and antibacterial test, SEM and AFM etc. The results well confirmed the hypothesis and behaviors provided in the introduction.
Point 2: The authors may want to verify if they meant ‘advantages’ or ‘disadvantages’ in line 41 of their introduction.
Response 2: The membrane fouling is adverse for the membrane performances, such as a shortness of service lifespan and the change of permeation and separation properties. In the revised paper, ‘…various advantages…’has been corrected into ‘…various disadvantages…’in line 42.
